# Direction-Based Hybrid Strategy Combining Pushing and Hitting for Fast Object Singulation

**Muhammad Umair Ahmad Khan [1], Sanghwa Kim [2] , Ji Yeong Lee [3],\* and Byung-Ju Yi [4],\***

1    Department of Biomedical Engineering, University of Engineering and Technology (UET) Lahore (Narowal) Campus, Narowal 51600, Pakistan; umairsiyal@gmail.com
2    Department of Electronics Engineering, Hanyang University, Ansan 15588, Korea; sonata7943@hanyang.ac.kr
3    Department of Robotics Engineering, Hanyang University, Ansan 15588, Korea
4    School of Electrical Engineering, Hanyang University, Ansan 15588, Korea
\*    Correspondence: jiyeongl@hanyang.ac.kr (J.Y.L.); bj@hanyang.ac.kr (B.-J.Y.); Tel.: +82-10-8819-7229 (J.Y.L.); +82-10-2065-5218 (B.-J.Y.)

**Abstract:** This paper presents a hybrid singulation strategy for fast object singulation in a cluttered environment. Recent techniques related to object singulation in clutter have employed various kinds of pushing techniques and in some cases have also used hitting techniques. However, these techniques have not addressed the issue related to the direction of pushing and hitting which is vital for fast object singulation. Finding the appropriate direction of hitting and pushing helps in singulating objects quickly in a cluttered environment. This paper proposes the desired direction for pushing and hitting, combined with a hybrid strategy, that results in fast object singulation in a cluttered environment. The number of times of pushing and hitting in terms of time is chosen as the measure of performance. We employ multiple circular disks as the test example and carry out diverse experiments to corroborate the usefulness of the proposed object singulation algorithm. This approach is able to singulate objects quickly in complex formations. In this paper, we have combined both pushing and hitting and also proposed the direction of hitting and pushing in order to singulate objects in clutter quickly.

**Keywords:** dynamic and quasi-static manipulation; object singulation; cluttered environment





## 1. Introduction

Humans pick and place things easily because humans are blessed with a brain that instantly finds out ways to complete the task. However, for robots, even the simple task is much more difficult and complex. There are two different ways to move an object from the initial position to the goal position by using a robot manipulator; prehensile and non-prehensile methods. In prehensile manipulation, the robot grasps the object and moves it to the target position. However, in non-prehensile manipulation, the robot moves the object to the target position by pushing and hitting. A combination of both techniques not only provides the desired object singulation but also consumes less time.

Manipulation methods can be divided into two types: quasi-static manipulation and dynamic manipulation. In the case of quasi-static manipulation, there is always contact between the manipulator and the object even after collision. Alternatively, in the case of dynamic manipulation, the object may lose its contact with the manipulator after collision.

Approaches related to robotic pushing have been extensively studied [1,2]. However, these approaches consume a lot of time since the pushing-only approach is helpful only for a few objects. In the case of multiple objects, this technique is not suitable. Dogar et al. [3] proposed a planning framework based on pushing in order to singulate an object in clutter. Although this approach helps in grasping objects that are ungraspable otherwise, the problem with this technique is the unnecessary pushing actions that result in the unnecessary displacement of objects.

As robots have already made their way into the daily lives of humans, they tend to perform things that are easy for humans. Pushing is one of the approaches utilized daily by humans in order to manipulate or grasp different objects. Stuber et al. [4] provided a literature review related to robotic pushing, where he has divided pushing into six major categories: (i) pure analytical, (ii) hybrid, (iii) dynamic analysis, (iv) physics engines, (v) data-driven, and (vi) deep learning.

In dynamic manipulation, the collision-based singulation approach is performed. Imran et al. [5] suggested an approach that was based on hitting. They have used a virtual world simulator that relied on a dynamic model and have estimated all of the physical properties of objects in an environment. After successful hitting in a virtual world, real-world hitting is performed. Different studies have also discussed approaches related to impulse-based techniques between a robot and an environment [6,7]. Moreover, it should be noted that, during the singulation by hitting, multiple impacts may occur simultaneously, and in that case, its analysis involves issues that cannot be solved by the simple extension of two-body impact, as discussed, for example, by Ivanov [8]. Smith et al. [9] and Raskit et al. [10] discussed the criterion that must be satisfied by the solution for simultaneous impacts and proposed methods for analyzing simultaneous impacts. These methods are focused on the analysis of the impact.

This paper has two contributions. First, we have proposed a hybrid approach that uses both pushing and hitting for the singulation of objects in clutter. Combining the advantages of pushing and hitting would be helpful in the singulation of objects in clutter. Second, we have also introduced the desired direction of hitting and pushing to facilitate the singulation of objects. Our approach based on the above contributes results in the fast singulation of objects in clutter. This paper proposes a blend of these two approaches in scenarios where only one approach alone cannot be effective in order to singulate objects. Section 2 discusses the motivation of this study. The hitting approach is introduced in Section 3. Sections 4 and 5 deal with a parameter estimation for a virtual world simulation and experimental verification. Section 6 deals with comparative analysis. Section 7 deals with discussion, and finally, we draw conclusions in Section 8.

## 2. Motivation and Strategy for Singulation

The hybrid approach, combining hitting and pushing, and the initial direction-based hitting and pushing, are the two main key ideas behind this study. Hitting is defined as a type of manipulation, where the object in contact with the manipulator keeps moving even after it loses its contact with the manipulator. Pushing is a type of manipulation, where there is always contact between the manipulator and the object. For pushing, the direction is determined from the common tangential line of two contacting disks, and for hitting the direction is chosen based on the 'central impact' which will maximize the impulse exerted on the bodies. These directions would be called 'the desired direction of object singulation' in this paper.

Impact occurs when bodies collide for a very short time during which a large impulse is exerted between the bodies, resulting in a large change of motion. The line passing through the contact point, and normal to common tangents of two bodies at the point of the contact, is known as the line of impact. In this paper, we consider the collision between the identical circular disks, and thus, the line passing through the centers of the colliding two disks is the line of impact.

Impact can be divided into two types of impact:

1. Central Impact;
2. Oblique Impact.

Central impact happens when the direction of motion of two colliding objects moving with velocity $\mathbf{v}_a$ and $\mathbf{v}_b$ is along the line of impact as shown in Figure 1a. Oblique impact happens when the direction of motion of one or both colliding objects are at some angle to the line of impact as shown in Figure 1b. It is also noted that more than two bodies may collide simultaneously. Multiple simultaneous impacts may occur as shown in Figure 1c.

As discussed in [10], such cases cannot be solved by the simple extension of two-body impact, and additional assumptions and/or constraints are needed. In this paper, since we are mainly interested in the scattering of the bodies which are initially at rest, we focus on the simultaneous impact where one moving body collides into two or more stationary bodies.

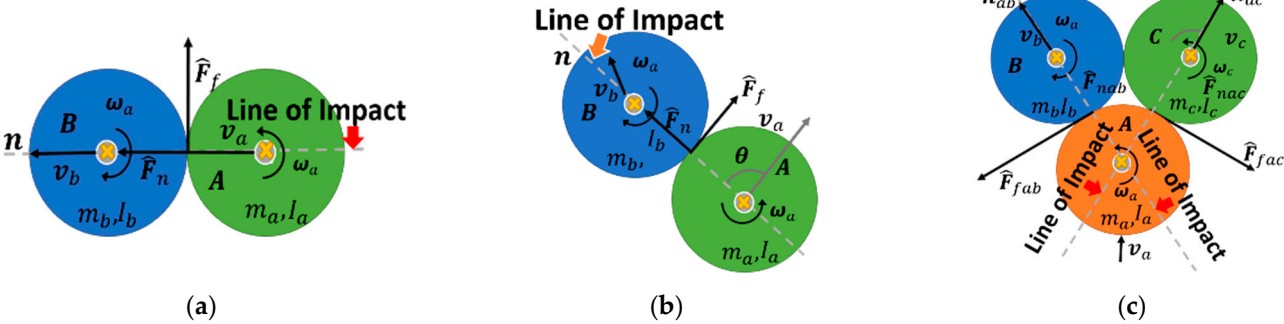

**(a)**                 **(b)**                 **(c)**

**Figure 1.** Impact types: (**a**) central impact contact, (**b**) oblique impact contact, and (**c**) multiple simultaneous impact contact.

*Choice of the Hitting/Pushing Directions for Singulation*

In this subsection, we describe the strategy of hitting and pushing to obtain singulation. Especially, we focus on the direction and the position of hitting or pushing. In the case of pushing, once the direction/position of pushing is chosen, the manipulator can separate the object by moving with any sufficiently slow velocity along a sufficiently large distance (e.g., in this paper, about the radius of the disk). That is the choice of the velocity, but it is not too important for the pushing. However, for hitting, the velocity should be carefully chosen. If the velocity is too small, impact will not occur. In the meanwhile, if the velocity is too big, the object may move outside the workspace of the robot. In this section, we discuss the choice of the direction/position of the hitting, and in Section 4, we discuss how to choose the velocity for hitting using a simulation.

As shown in Figure 1, the impulse can be resolved into a tangential component $\hat{F}_f$ and normal component $\hat{F}_n$. In this study, we assume negligible friction between colliding bodies. The tangential component of the impulse $\hat{F}_f$ is due to the friction between two bodies, and if we assume that the friction between the bodies is negligible, we only need to consider the normal component $\hat{F}_n$ of the impulse.

The desired initial direction for pushing for any two circular contacting disks is along the common tangential line of the two disks as shown in Figure 2. This is called central pushing. This is desirable because it dissipates minimal energy compared to pushing a circular disk along its center, as shown by the gray arrows in Figure 3, which dissipates high energy. Alternatively, the initial direction for hitting has been selected on the basis of central impact which will maximize the impulse. It can be simply observed from Figure 3.

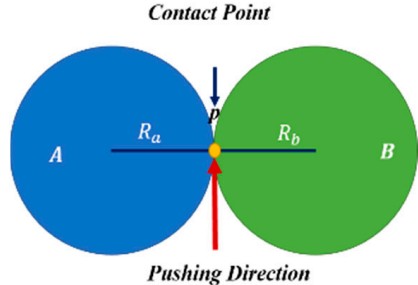

**Figure 2.** Pushing direction for two contacting circular disks.

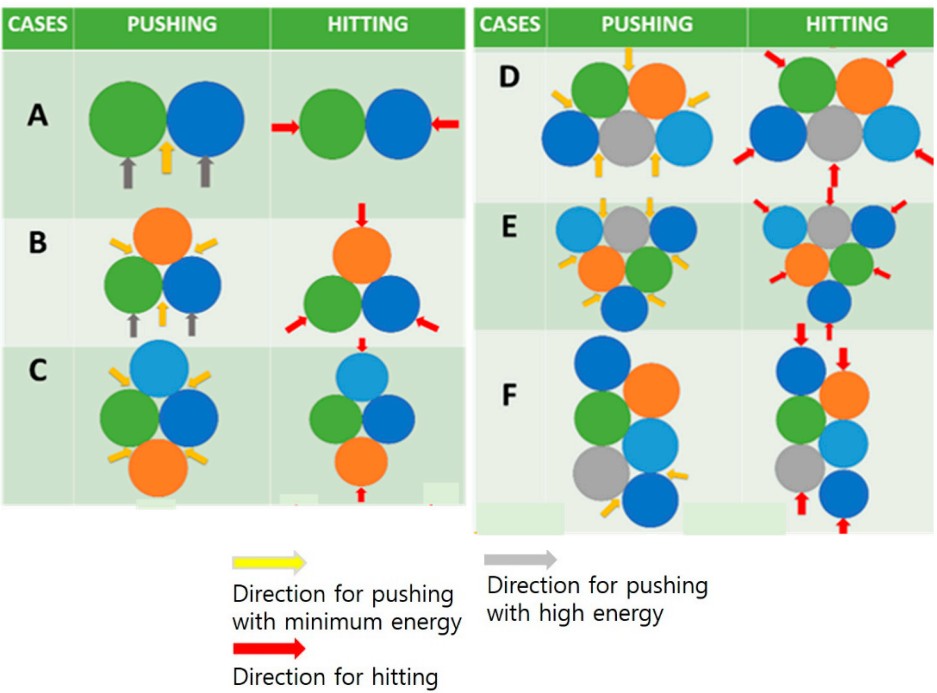

**Figure 3.** Pushing vs. Hitting directions: (**A**) two circular discs in symmetric formation (**B**) three circular discs in symmetric formation (**C**) four circular discs in symmetric formation (**D**) five circular discs in symmetric formation (**E**) six circular discs in symmetric formation (**F**) six circular discs in asymmetric formation.

that the directions of pushing and hitting are orthogonal to each other in the sense that the pushing is trying to minimize energy for the singulation of objects, while the hitting is trying to maximize energy for the singulation of objects.

The yellow and gray arrows show the directions for pushing and the red arrow shows the direction for hitting. The pushing along the yellow arrow direction will need less energy because we are just pushing the manipulator from below. For the grey arrow, we may need to push either the green or blue disk for Case A all the way up, approximately the distance equal to the radius of the disk. As a result, it will consume more energy. Figure 3 shows multiple arrows for different cases and these arrows represent the direction of hitting and pushing with respect to the manipulator. For example, in Case C if the orange circular disk is closer to the manipulator, the red arrow shows the hitting direction. Similarly, the two yellow arrows show the pushing directions. In this case, the manipulator will always choose the pushing direction arrow nearer to the manipulator.

The central pushing approach of two contacting circular disks reduces the possibility of collision with other disks in the clutter, since this approach moves the disks because it only singulates by creating the minimum separation required for the desired singulation and also reduces the possibility of going beyond the workspace. This approach also helps in separating the disks at the point of contact without dragging them along other disks. Given the two disks in contact, the pushing direction is along the common tangent of the two disks at the point of contact. These two disks in Figure 2 can be separated easily by moving the manipulator by the fixed distance, along the direction shown by the arrow. The distance depends on the radius of the disks.

In the case of pushing, Case A and B of Figure 3 can be singulated by using a single push, but Case C requires two pushes. Case D, E, and F need multiple pushes. It can be easily observed that the number of pushes increases as the number of parts in the clutter increases. For hitting, the direction of hitting is orthogonal to that of pushing as shown in the first row of Figure 3. This direction applies high energy to singulate two objects. Cases from A to E in Figure 3 have symmetric formations related to billiards and can be singulated using a single hit. In order to minimize the singulation time or effort, hitting is

suitable when the number of circular disks is more than four. However, for Case F, having asymmetric formation, a single hit cannot singulate that formation and as a result, we need to use pushing after hitting in order to singulate all of the circular disks. This is the motivation of the hybrid approach combining hitting and pushing proposed in this study.

### 3. Hitting Approach

In the case of hitting, we have considered two types of circular disk formations as examples:

(a)    Billiard Formation;
(b)    Arbitrary Formation.

In the case of billiard formation, as shown in Figure 4, circular disks are increased in rows one by one, starting from one disk in row 1, two disks in row 2, three disks in row three, and so on. We have termed these rows as levels. All of the circular disks are tightly placed together and are contacting each other. When the manipulator hits the yellow circular disk at level 1, this impulse is sequentially propagated through all of the levels until it reaches the last level 3.

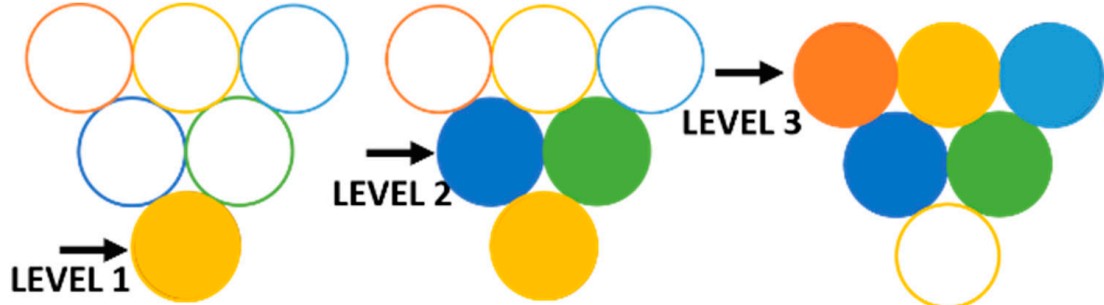

**Figure 4.** Impulse propagation among billiard balls.

Figure 4 shows the sequential propagation where empty and full circular discs are used to depict this phenomenon. When the manipulator hits the yellow circular disc at level 1, the impulse from the manipulator is transferred to level 1 and the full yellow circle represents that. Similarly, all of the above circular discs are empty which shows that the impulse is still not propagated to them. In the same way, the impulse is propagated to level 2, and finally, it will propagate to level 3. We analyze the propagation of the impulse level by level, considering only two levels at one time. The underlined assumption of the sequential collision process is that there exists some time delay between the levels. This is due to the microscopic deflection of two colliding disks when collision happens [10].

Assumptions are given as follows:

1.    Neglect friction between disks;
2.    Collision propagates in a sequential manner.

### 3.1. Impact Analysis for Billiard Formation

Let $m$ be the mass of the discs (which are identical and thus have the same masses), and $e$ be the coefficient of restitution between any two discs among the discs $i$, $j$, $k$ (again, the coefficients of restitution between any two discs are the same since the discs are identical). Let $n_{ij}$, $n_{ik}$ be the common normal of the discs $i$ and $j$, and of the discs $i$ and $k$, respectively. $\mathbf{v}_i$ and $\mathbf{v}'_i$ are the velocities of disc $i$ before and after collision, respectively. $\mathbf{v}_j$, $\mathbf{v}'_j$, $\mathbf{v}_k$, and $\mathbf{v}_k\prime$ are the velocities of the discs $j$ and $k$ before and after collision.

First, the total momentum is conserved, that is:

$$m\mathbf{v}_i + m\mathbf{v}_j + m\mathbf{v}_k = m\mathbf{v}'_i + m\mathbf{v}'_j + m\mathbf{v}'_k. \tag{1}$$

which is a 2D vector equation. Thus, two scalar equations exist. Since we ignore the friction between the discs during collision, the tangential components of the momenta of discs $j$ and $k$ are conserved, that is:

$$\left(m\mathbf{v}'_j\right)_{t_{ij}} = \left(m\mathbf{v}_j\right)_{t_{ij}}, \left(m\mathbf{v}'_k\right)_{t_{ik}} = \left(m\mathbf{v}_k\right)_{t_{ik}}, \tag{2}$$

thus,

$$\left(\mathbf{v}'_j\right)_{t_{ij}} = \left(\mathbf{v}_j\right)_{t_{ij}}, \left(\mathbf{v}'_k\right)_{t_{ik}} = \left(\mathbf{v}_k\right)_{t_{ik}} \tag{3}$$

where the subscript $t_{ij}$, $t_{ik}$ denotes the components of the velocities along the respective tangential directions (that is, the directions normal to the common normal $n_{ij}$, $n_{ik}$, respectively). These are two scalar equations.

From the impact equation, we have:

$$e = \frac{\left(\mathbf{v}'_i - \mathbf{v}'_j\right)_{n_{ij}}}{\left(\mathbf{v}_j - \mathbf{v}_i\right)_{n_{ij}}}, \ e = \frac{\left(\mathbf{v}'_i - \mathbf{v}'_k\right)_{n_{ik}}}{\left(\mathbf{v}_k - \mathbf{v}_i\right)_{n_{ik}}} \tag{4}$$

which give two scalar equations. Thus, we have six scalar equations, enough to solve for six unknowns (that is, $\mathbf{v}'_i$, $\mathbf{v}'_j$, $\mathbf{v}'_k$).

Using the coordinate system as shown in Figure 5 (in which the $x$-axis is parallel to the line connecting the centers of the discs $j$ and $k$, such that the angle of $n_{ij}$ and $n_{ik}$ are 60° and 120° from the $x$-axis, respectively), the tangential and normal velocity components of three objects can be decomposed into x and y directions as follows:

$$v_{jn_{ij}} = v_{jx} \cos 120 + v_{jy} \sin 120 \tag{5}$$

$$v_{jt_{ij}} = -v_{jx} \sin 120 + v_{jy} \cos 120 \tag{6}$$

$$v_{kn_{ik}} = v_{kx} \cos 60 + v_{ky} \sin 60 \tag{7}$$

$$v_{kt_{13}} = -v_{kx} \sin 60 + v_{ky} \cos 60 \tag{8}$$

$$v_{in_{ij}} = v_{ix} \cos 60 + v_{iy} \sin 120 \tag{9}$$

$$v_{in_{ik}} = v_{ix} \cos 120 + v_{iy} \sin 60 \tag{10}$$

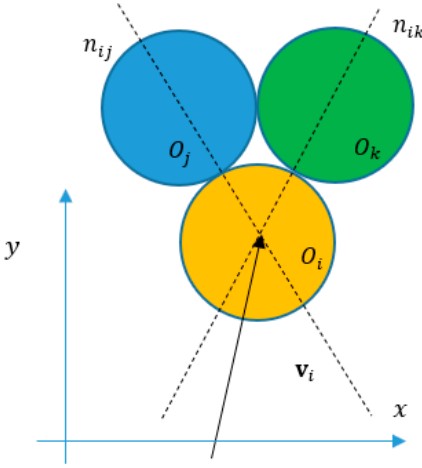

**Figure 5.** Simultaneous impact model.

Substituting Equation (5) through Equation (10) then into Equations (1)–(3), we can find the velocity vectors $\mathbf{v}'_i$, $\mathbf{v}'_j$, $\mathbf{v}'_k$ of the three discs in the $xy$ coordinate as follows:

$$v_i' = \frac{1}{3}(2-e)v_{ix}i + \frac{1}{5}(2-3e)v_{iy}j \tag{11}$$

$$v_j' = \frac{1}{30}\left(5(1+e)v_{ix} - 3\sqrt{3}(1+e)v_{iy}\right)i$$
$$+ \frac{1}{30}\left(-5\sqrt{3}(1+e)v_{ix} + 9(1+e)v_{iy}\right)j \tag{12}$$

$$v_k' = \frac{1}{30}\left(5(1+e)v_{ix} + 3\sqrt{3}(1+e)v_{iy}\right)i$$
$$+ \frac{1}{30}\left(5\sqrt{3}(1+e)v_{ix} + 9(1+e)v_{iy}\right)j \tag{13}$$

The impact analysis between levels 1 and 2 can be generalized to the impact between the levels *n* and *n* + 1 in the following manner. Consider the motion of the disks in level 3 influenced by the motion of the disks in level 2. We assume that, as shown in Figure 6, disk 5 is hit by disks 2 and 3 at the same time. We compute the velocity of disk 5 after the impact, first by analyzing impact between disks 2, 4, and 5, and impact between disks 3, 5, and 6 using Equations (4)–(6) as above, and then by adding the velocities obtained for disk 5 from these two impacts. Without the loss of any generality, we will consider a group of three disks in which one moving disk is colliding into two stationary disks, and if a stationary disk is hit by two disks at the same time, we add the result from two analyses to get the velocity of that disk after the collision. This process will be continued from level *i* to level *i* + 1, where *i* ranges from 2 to the final level.

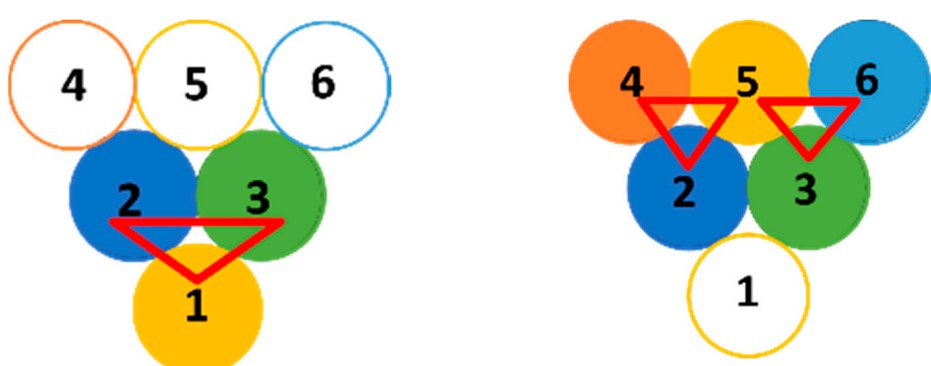

**Figure 6.** Triangular approach to calculate the velocities after impact.

*3.2. Impact Modeling between Two Bodies*

When the robot manipulator collides with circular disk A, then circular disk A will collide with disk B in the environment. Assuming the central impact, the friction between two colliding bodies is negligible. Considering that two circular disks with mass center velocities $v_a$, $v_b$ are approaching and contacting, then the dynamic model of each disk can be described as:

$$F_{ext} = m_a \dot{v}_a \overset{\int \cdot dt}{\to} \hat{F}_{ext} = m_a(v_a' - v_a) \tag{14}$$

$$-F_{ext} = m_b \dot{v}_b \overset{\int \cdot dt}{\to} -\hat{F}_{ext} = m_b(v_b' - v_b) \tag{15}$$

where $m_a$ and $m_b$ denote the mass of body 'A' and body 'B', respectively. Rewriting Equations (7) and (8), we have:

$$\Delta v_a = \hat{F}_{ext} m_a^{-1} \tag{16}$$

$$\Delta v_b = \hat{F}_{ext} m_b^{-1} \tag{17}$$

where $\Delta v_a = \overline{v}_a' - \overline{v}_a$ and $\Delta v_b = \overline{v}_b' - \overline{v}_b$. $\overline{v}_a'$ and $\overline{v}_b'$ denotes the post-impact velocity vectors of body 'A' and body 'B', respectively. The closed-form solution of the external impulse is obtained as follows:

$$\hat{F}_{ext} = \frac{-(1+e)(\overline{v}_a - \overline{v}_b) \cdot \boldsymbol{n}}{m_a^{-1} + m_b^{-1}}. \tag{18}$$

The above analytical derivation will be used to estimate the motion of all objects just after collision in a virtual world simulation.

We can write the following relation where $v_I'$ and $v_a'(t_o)$ are the velocity vector of the end effector of a robot after collision and the velocity of disk A after the collision. In the case of hitting, the velocity $(v_a'(t_o))$ of the disk is greater than the velocity $(v_I')$ of the robot manipulator:

$$v_I' < v_a'(t_o), \tag{19}$$

while in the case of pushing, we can write the following relationship:

$$v_I' = v_a'(t_o) \tag{20}$$

which implies that in the case of pushing, the robot moves with disk A.

## 4. Virtual View Simulations for Hitting

Virtual view simulations are performed in Matlab in order to create a virtual world for hitting that is based on the estimated parameters. To get the desired singulation in the real world, virtual view simulations are carried out using Matlab. Iterative singulations are performed to achieve the desired distance among disks using virtual view simulations by providing the initial formation and physical properties of circular disks to the virtual world. When the desired singulation has the minimum desired distance among disks, achieved by using the iterative approach in a virtual world at a specific velocity, then the same velocity is fed to the real world manipulator to achieve the desired singulation in the real world. Given a formation, the hitting direction is chosen as discussed in Section 2 which is based on either central or simultaneous impact contact between circular disks. We determine the hitting velocity through simulation, implemented by Matlab. In the simulation, we repeatedly hit the formation with the chosen direction, while increasing velocity until the desired distance is obtained. In symmetric billiard formation, the desired distance between six disks can be achieved by using the iterative approach. However, for asymmetric formations of six circular disks, the desired distance among all of the disks cannot be achieved by using only hitting so the iterative singulation in a virtual world is performed until the disks remain inside the predefined workspace, and all of the disks which are still contacting each other, even after hitting, are singulated by using the pushing approach.

## 5. Experimental Verification

This section deals with experimentation related to both pushing and hitting. We have used an Intel RealSense D435i Depth Camera and applied the image segmentation technique to see whether the objects are in contact with each other or not. The RGB image is converted to HSV color space because HSV color space is less affected by brightness compared to RGB. As shown in Figure 7, the circular disks are segmented according to the HSV range. As an example, if the circular disk is red, the area is set to 255 and in all other cases, it is set as 0. The results of segmentation for each color are stored. To remove noise, erosion is performed immediately after dilation and the image is returned to the original image size. As a result of each segmentation, the contour of the image is obtained. The image contour is defined as the line formed when circular disks contact each other. The thickness of each contour is made thicker so that contours can overlap when there exist adjacent circular disks. To find the overlapping between two adjacent circular disks, each image of the circular disk is compared. The contour has a start point and an end point. The nearest point of contour facing the manipulator is selected as the start point and the end point is the point where the contour finishes applying Principal Component Analysis (PCA) to the region of the overlapping contour. Rows and columns constitute 2D image

coordinates. In a 2D binary image, two vectors are used to represents the 255 intensity region, where 255 changed to 1 represents white, while black is represented by 0. These two vectors represent the longest and shortest axes and the direction of pushing is chosen along the longest axis. The length of the pushing stoke is a combination of the length of contour and the length of the end effector such that the manipulator does not stop at the end point of the contour and moves further equal to the length of the end effector to singulate the contacting disks.

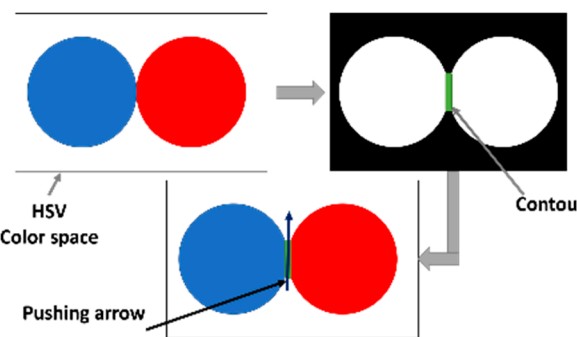

**Figure 7.** Image segmentation for pushing.

Figures 8 and 9 show the cases of single pushing, where two and three circular disks contact each other. Initially, a vision algorithm based on PCA-based image segmentation is used to create a contour and pushing arrow. Resultantly, the robot will push along the arrow in order to singulate the circular disks contacting together. After the first push, the manipulator again uses the RealSense camera view to see any other contour. It is found that for both two and three circular disk cases as shown in Figures 8 and 9, one push is able to yield the desired singulation.

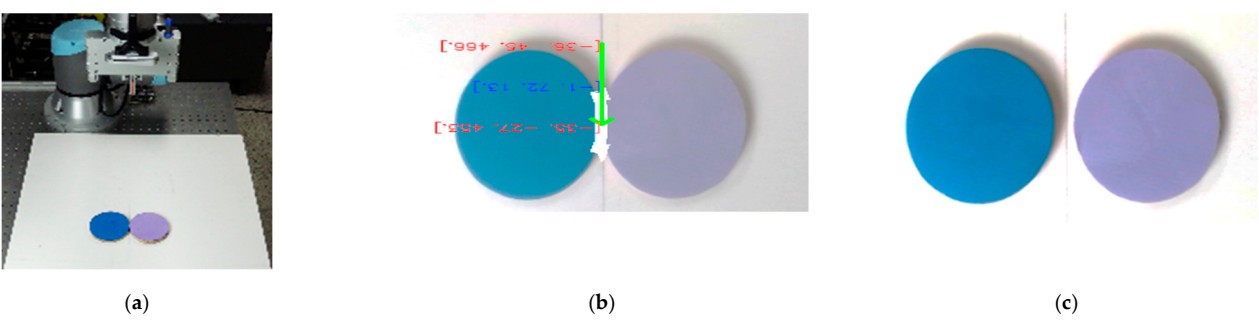

(a)  (b)  (c)

**Figure 8.** Two circular disk formation: (**a**) experimental setup, (**b**) RealSense camera view showing the contour and the direction of pushing arrow, and (**c**) RealSense camera view showing desired singulation.

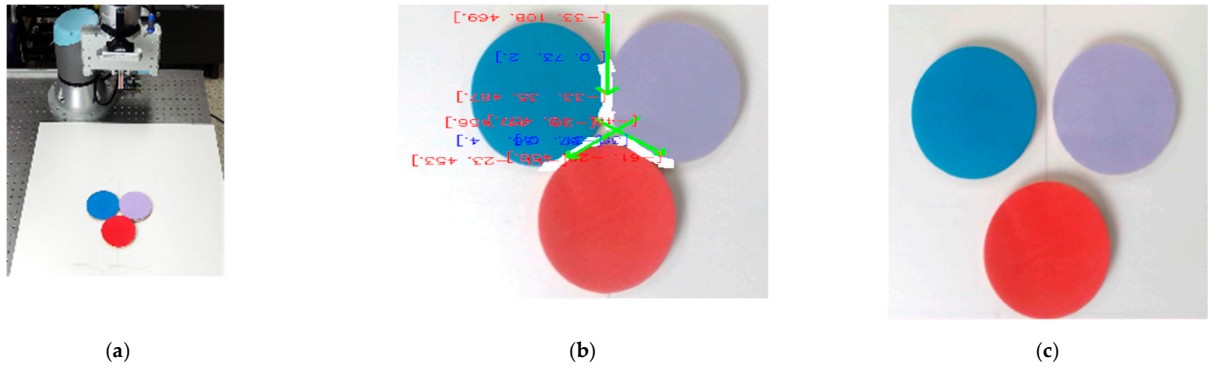

(a)  (b)  (c)

**Figure 9.** Three disk formation: (**a**) experimental setup, (**b**) RealSense camera view showing the direction of pushing, and (**c**) camera view after singulation.

In Figure 9, the RealSense camera view shows three arrows. The green arrow that is between the blue and purple disks is chosen as the pushing arrow since it lies closer to the end effector of the manipulator. Figure 10 shows the four circular disk formation and as explained earlier the vision algorithm will be used to predict the pushing directions and in the case of four disks, it will require two pushes in order to get the desired singulation among the disks. First, the robot tries to singulate them by the upper-left arrow between the blue and red disks. The four circular disks can be singulated using two pushes.

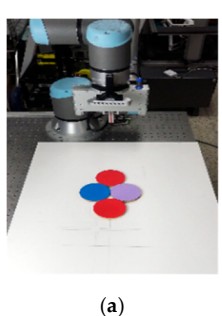 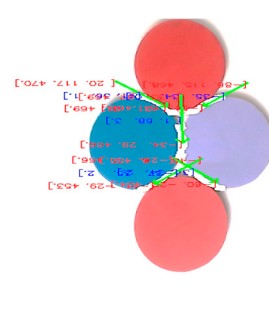 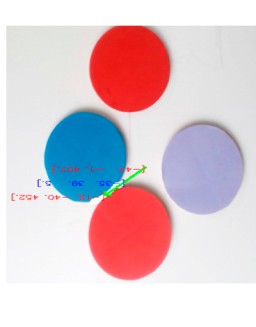 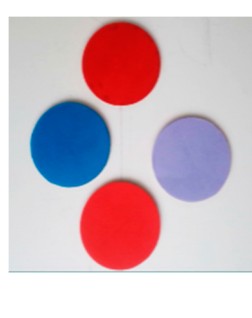

(**a**) (**b**) (**c**) (**d**)

**Figure 10.** Four circular disk formation: (**a**) experimental setup, (**b**) RealSense camera view showing the direction of pushing arrow (**c**) experiment result after the first push, and (**d**) experiment result after the second push.

Figure 11 shows the billiard formation of six discs. The position of the first circular disk is set a little far from the other five disks so that it acts as a cue disk like a cue ball in billiards. The manipulator will hit the blue disk and then the blue disk will hit two disks in level 2.

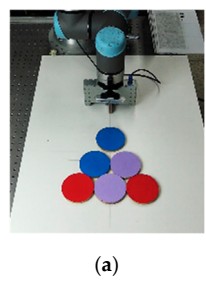 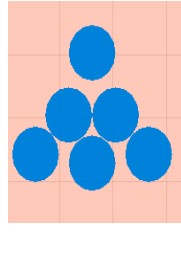 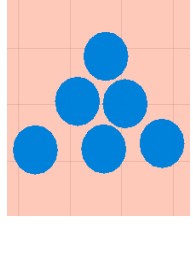 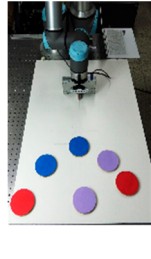

(**a**) (**b**) (**c**) (**d**)

**Figure 11.** Six circular disks billiard formation: (**a**) initial formation of six circular disk billiard formation in real-world, (**b**) initial formation of six circular disk billiard formation in virtual-world, (**c**) desired singulation using single hitting in virtual world showing all disks are singulated, and (**d**) desired singulation using a single hit in real-world showing all disks are singulated.

Then it will be propagated to level 3. As discussed earlier, if we use only pushing, it needs multiple pushes to separate this formation, which is time-consuming. Before conducting the real experiment, we have performed hitting virtually using the Matlab simulation tool to estimate an appropriate robot velocity for the singulation of all of the objects. Figure 11c,d shows both the virtual view and the real view of singulation, and only one hit can result in the desired singulation. Here, the desired singulation implies that all of the objects are separated enough to grasp them with a gripper without collision.

Figure 12 shows the asymmetrically arranged six circular disk formation. It is found that after hitting in the virtual simulator, few circular disks are still in contact as shown in Figure 12c. After hitting along the red arrow, we require additional pushing to complete the singulation of all of the objects. The RealSense view in Figure 12e shows purple and red disks contacting each other and as a result, there is a contour between the two disks with the pushing direction arrow. The first push created a new contacting body between red and blue as shown in Figure 12f. So, again, a contour and pushing arrow appears between the two circular disks that indicate the second pushing. Figure 12g shows desired singulation.

As a result, such an asymmetric formation achieves the desired singulation using one hit and two pushes. Figure 12e shows a small contour between the red and blue disks although these disks are not in contact with each other, this was because of the lighting issue. Figure 13 shows another asymmetrically arranged six circular disk formation and the hitting direction is chosen on the basis of simultaneous impact. After hitting two circular disks, the red and blue are still contacting each other as shown in Figure 13d. Followed by additional pushing, the desired singulation is achieved as shown in Figure 13e.

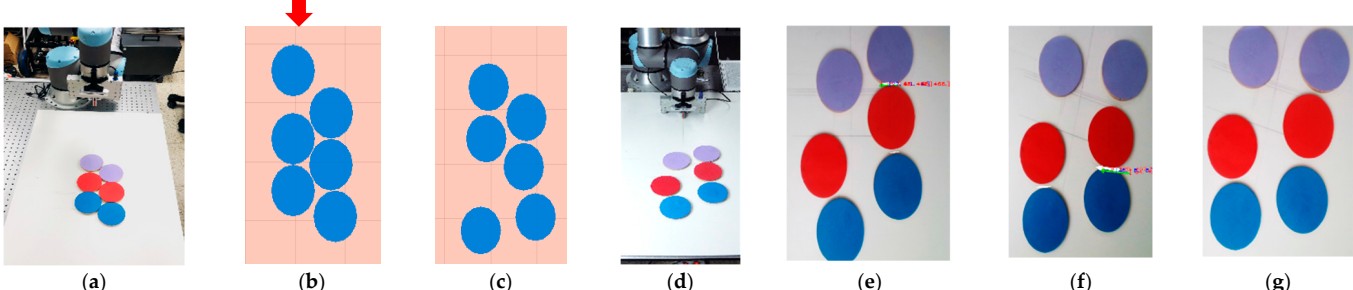

| (a) | (b) | (c) | (d) | (e) | (f) | (g) |

**Figure 12.** Six circular disks in asymmetric formation: (**a**) experimental setup, (**b**) virtual view of six circular disks: initial asymmetric formation, (**c**) virtual view after single hit, (**d**) real view after single hit, (**e**) RealSense camera view showing a contour and pushing direction green arrow between contacting purple and red disks, (**f**) camera view after first push showing a contour and pushing direction by green arrow between contacting red and blue disks, and (**g**) camera view.

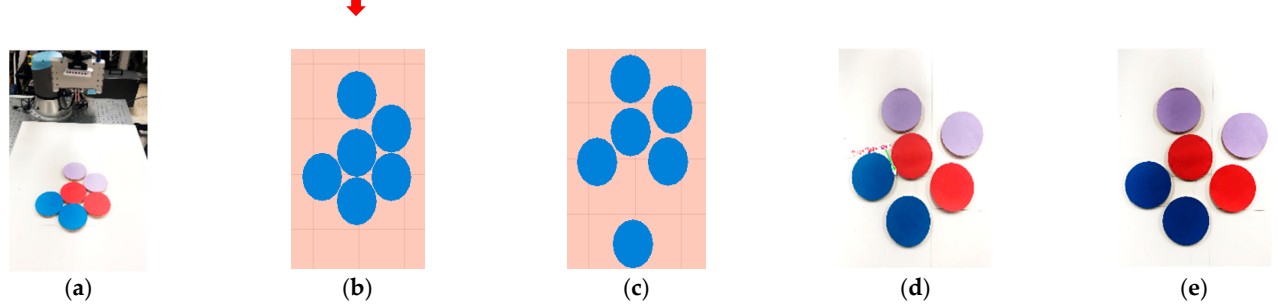

| (a) | (b) | (c) | (d) | (e) |

**Figure 13.** Six circular disks in asymmetric formation: (**a**) experimental setup, (**b**) virtual view showing initial asymmetric formation, (**c**) Virtual view of six circular disks after single hit, (**d**) RealSense camera view showing a contour and pushing direction by green arrow between contacting blue and red disks, and (**e**) RealSense camera view after first push shows the desired singulation.

The result from the simulator and the result from the experiment are not always the same, as shown in Figures 11–13, because the virtual view simulations are done on the basis of estimated physical parameters. This is due to the difference between the true and the estimated values of the parameters. Especially, in our experimental setup, the friction was not uniform over the surface, while in the simulation the coefficient friction was assumed to be constant.

## 6. Comparative Analysis among Initial Desired Direction-Based Hybrid Singulation and Other Random Singulation Approaches

This section describes the comparative analysis of the three approaches: initial desired direction-based hybrid singulation approach and random hybrid singulation (direction of pushing and hitting is random), and random pure pushing approach (only pushing technique is used for singulation). To validate the effectiveness of our initial direction-based approach, we performed three experiments using six circular disks in different configurations with the initial direction-based hybrid approach, the random hybrid approach, and the random pure pushing approach and measured the time it took for complete singulation. It took the manipulator on average 4.5 s for our proposed direction-based hybrid singula-

tion, 11.16 s for the random hybrid singulation, and 24.5 s for the random pure pushing approach. Figure 14 shows the comparison with respect to time in three cases. Figure 14 shows the effectiveness of our proposed approach in the singulation of objects in clutter in terms of the time taken by each approach.

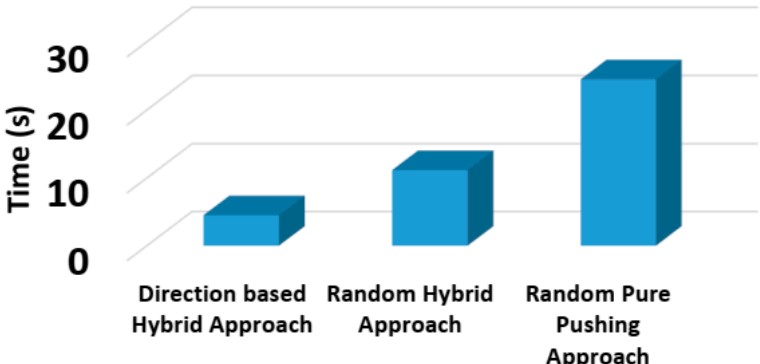

**Figure 14.** Comparison among our proposed direction-based hybrid singulation approach and other approaches.

## 7. Discussion

In this study, we proposed the initial direction-based hybrid singulation approach. In the start, we used hitting to scatter the objects to achieve the singulation of disks in clutter. All of the disks were not singulated using just a single hit, as a result, we needed to use pushing to completely separate all of the objects. Previous studies have used pushing to singulate objects but they have not developed a physics-based analytical model to find the appropriate direction of pushing or hitting. In our study, our focus was to reduce the overall time taken in order to singulate objects effectively.

Several studies based on the impulse between the robot and the environment have been discussed [6,11]. However, dynamic and quasi-static approaches alone cannot achieve singulation for various cases. Papallas et al. studied a non-prehensile manipulation approach in clutter using a human [12]. However, this approach needs an input from a human which consumes time. In this approach, they have used pushing as a non-prehensile manipulation method to manipulate objects in clutter. In some studies, there is the problem of approaching invisible spaces which creates a limitation for the system to singulate objects in clutter [13]. However, this problem did not arise in this study and all of the disks were visible in the workspace and there were no such invisible areas.

Several studies have faced the problem of the unnecessary displacement of objects [3]. The main reason for this problem is the randomized hitting or pushing that cause the unnecessary displacement of objects. Our approach, based on the initial direction for hitting and pushing, solved this problem because it helped the disks to remain in the workspace when relying on a hybrid approach of hitting and pushing.

In this paper, the main focus was to perform a hybrid singulation using the initial direction based on our mathematical model and the emphasis was on acquiring the desired singulation among circular disks in less time. In the future, we would like to extend this study to singulate objects of various sizes and shapes.

Although this paper has focused on an analytical way of object singulation in clutter, a learning-based approach has many benefits. Reinforcement learning-based pushing for obtaining optimal pushing policies is another way proposed to singulate objects in clutter. However, this approach lacks pushing in more directions [14]. In various scenarios where it is hard to find grasps, pushing or sweeping actions can be used for automated bin picking to singulate objects. Eitel et al. [15] proposed a novel neural network-based approach that separates unknown objects in clutter by selecting favorable push actions. Chang et al. [16]

considered the singulation of objects by alternating pushing and grasping. Here, pushing was used as a test action to evaluate whether an isolated set of cloud points is a single object or not. If the set of cloud points splits into multiple units after pushing, it repeats the pushing for these smaller units. If not, the robot tries to grasp the object (represented as a set of cloud points). A heuristic for choosing the direction and length of pushing was given purely based on the geometry of the cloud, without considering the dynamic property of the object. Danelczuk et al. [17] compared five pushing policies based on object separation and grasp confidence, which is probability of grasp success. For each pushing strategy, the degree of object separation and grasp confidence was computed and it is shown that the degree of object separation is not necessarily correlated with grasp success. Marios et al. [14] used a learning-based method to singulate objects. They did not explicitly consider the geometry of the object and also use a limited number of pushing directions, which can increase the number of pushing.

In the future, incorporating a learning method into an analytical method would enhance the overall performance of object singulation in clutter.

## 8. Conclusions

The contribution of this paper is the proposition of a new initial direction-based hitting and pushing-based hybrid approach in order to singulate objects faster. It was analyzed that the directions of pushing and hitting are orthogonal to each other in the sense that the pushing is trying to minimize energy for the singulation of objects, while the hitting is trying to maximize energy for the singulation of objects. Though the maximum six circular disks, both in the symmetric billiard and asymmetric formations, are treated as examples, it can be extended to a general formation with multiple levels. This approach in comparison with previous approaches has two main advantages: (i) shorter singulation time and (ii) the absence of unwanted clutter motion due to the retraction of the manipulator. An analytical model of the impulse between two colliding bodies was used to estimate the required velocity of the robot manipulator to obtain the singulation of all of the objects. For hitting, the central impact model was used and the simultaneous central impact approach was also used to propagate the impulse from the current level to the next level. We verified the effectiveness of the proposed hybrid singulation algorithm through experimentation.

It is noted that the results presented in this paper are promising but preliminary, and more experiments are needed to provide a reliable conclusion. In the future, we would like to extend this study for irregular-shaped objects and different stacking cases, where multiple objects stacked in clutter would be singulated.

**Author Contributions:** Conceptualization: B.-J.Y., M.U.A.K. and J.Y.L.; methodology: M.U.A.K. and B.-J.Y.; software: M.U.A.K. and S.K.; writing—original draft preparation: M.U.A.K.; writing—review and editing: M.U.A.K., J.Y.L. and B.-J.Y. supervision: B.-J.Y. and J.Y.L.; experimentation: S.K. and M.U.A.K. All authors have read and agreed to the published version of the manuscript.

**Funding:** This research was supported by the Technology Innovation Program (grant no: 20001856, Development of Robotic Work Control Technology Capable of Grasping and Manipulating Various Objects in Everyday Life Environment Based on Multimodal Recognition and Using Tools) funded by the Ministry of Trade, Industry and Energy (MOTIE, Korea) and was supported by BK21 FOUR (Fostering Outstanding Universities for Research) funded by National Research Foundation of Korea (NRF).

**Conflicts of Interest:** The authors declare no conflict of interest.

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
