# Peer review of "Direction-Based Hybrid Strategy Combining Pushing and Hitting for Fast Object Singulation"

_applsci, doi:10.3390/app11167327_

Round 1

Reviewer 1 Report

This paper deals with the problem of singulating identical circular disks in a scene by employing non-prehensile manipulation actions. Specifically, the authors utilize push and hit actions in order to achieve singulation.

The pushing direction is determined from the common tangential line of two contacting disks. Since the direction is chosen, the robot moves with sufficiently slow velocity for a distance defined by the radius of the disks.

As far as the hitting primitive is concerned, the direction is chosen based on the 'central impact' which will maximize the impulse exterted to the bodies. The authors make the assumption that the friction between the bodies is negligible and as a result only the normal force of the impulse between the objects needs to be computed.

They model the collision between the circular disks in order to determine the appropriate hitting velocity of the robot so as the disks will be singulated and not move outside of the robot's workspace. Specifically, they compute the exterted forces to each object after a collision with other objects (simultaneous impacts between circular disks are considered). The appropriate hitting velocity is determined through simulation experiments after multiple runs. Given the formation of the disks and the selected direction, they repeatedly execute the hitting primitive, while increasing the velocity until the desired singulation distance is obtained.

Overall, the paper is not well written and it is very difficult to follow. There is not clear which are the contributions of the proposed method. It seems that the only contribution is the hitting primitive which makes unrealistic assumptions about the friction of the objects. The authors are desired to rewrite the contributions in order to address their uniqueness. Furthermore, the authors perform experiments only with circular disks, but not with any other type of objects. It would be very helpful to conduct experiments with a more diverse set of objects in order to evaluate the utility of the hitting primitive.

Questions / Concerns
1) The authors make the assumption that the friction between the bodies is negligible which may not state in a real environment. Furthermore, they do not take into account the friction between the objects and the planar surface which cloud lead to very different results.
2) How do you end up in the equations 4, 5 and 6?
3) There is no comparison with other singulation methods as [4].

Minor Concerns
1) I think that the [4] citation is incorrect. The correct one is 'M. Kiatos and S. Malassiotis, "Robust object grasping in clutter via singulation," 2019 International Conference on Robotics and Automation (ICRA), Montreal, QC, Canada, 2019, pp. 1596-1600, doi: 10.1109/ICRA.2019.8793972.'

Reviewer 2 Report

This paper proposes the initial direction for pushing and hitting, combined with hybrid strategy that results in fast object singulation in cluttered environment.

The authors employ multiple circular disks and carry out different experiments to corroborate the usefulness of the proposed object singulation algorithm, in order to to singulate objects quickly in complex formations.

The paper is interesting and well written.

However, I suggest following the suggestions above, in order to further enhance it.
In particular, the authors should focus more on conclusions.

Furthermore, I suggest improving the analysis of literature.

I encourage the authors to refine their work to make it available for publication in the journal.

Author Response

Reviewer2

[Comment 2-1]

 The authors employ multiple circular disks and carry out different experiments to corroborate the usefulness of the proposed object singulation algorithm, in order to to singulate objects quickly in complex formations. The paper is interesting and well written. However, I suggest following the suggestions above, in order to further enhance it. In particular, the authors should focus more on conclusions Furthermore, I suggest improving the analysis of literature. I encourage the authors to refine their work to make it available for publication in the journal.

[Answer 2-1]

Thank you for pointing it out. Both conclusion section and analysis of literature have been improved by adding following passages.

Conclusion

“This approach in comparison with previous approaches has two main advantages: (i) shorter singulation time (ii) absence of unwanted clutter motion due to the retraction of manipulator”.

Analysis of literature

“A pushing based neural network approach was utilized to singulate unknown objects in clutter [5]. However, in various scenarios where it is hard to find grasps, pushing or sweeping actions can be used for automated bin picking to singulate objects [6]. Reinforcement learning based pushing and grasping based on image morphological operations were used to manipulate the objects [7-8]. The above mentioned simplified pushing approach led to longer singulation time and unwanted clutter motion due to the retraction of manipulator”.

Reviewer 3 Report

The results are up to my knowledge correct. They are well presented but some minor language mistakes must be corrected. 

The novelty of the paper is lacking - or not clearly presented.

In my opinion the authors should clearly present what is the main contribution of their approach when compared to others in the literature.

Author Response

Reviewer3

[Comment 3-1]

 The results are up to my knowledge correct. They are well presented but some minor language mistakes must be corrected. The novelty of the paper is lacking - or not clearly presented.In my opinion the authors should clearly present what is the main contribution of their approach when compared to others in the literature.

[Answer 3-1]

Thank you for pointing it out. Minor language mistakes are corrected and highlighted. The contribution of the paper is clearly highlighted in the updated version of the paper by adding following passages

“This paper has two contributions. First, we have proposed a hybrid approach that uses both pushing and hitting for singulation of objects in clutter. Combining the advantages of pushing and hitting would be helpful in singualtion of objects in clutter. Second, we have also introduced the initial direction of hitting and pushing. Our approach based on above contributions, results in fast singulation of objects in clutter”.

“This approach in comparison with previous approaches has two main advantages: (i) shorter singulation time (ii) absence of unwanted clutter motion due to the retraction of manipulator”.

Round 2

Reviewer 1 Report

In this paper, the authors introduce an approach that combines two non-prenhesile actions, pushing and hitting, in order to singulate identical circular disks. In the revised manuscript, the authors did not address the reviewer's concerns.

1) The paper is still not well written.

2) The authors claim that their second contribution is the proposal of 'the initial direction of hitting and pushing'. This needs to be clarified.

3) The equations 4,5 and 6 are still confusing. It is not clear how the authors end up in these equations.

4) There is still missing qualitative comparison of the proposed approach with learning
([1]) or analytic baselines ([2], [3]). Both analytic baselines employ pushing actions and thus they could be trivially adapted to the circular disk scenario.

5) Experiments with a more diverse set of objects are not conducted.

[1] M. Kiatos and S. Malassiotis, "Robust object grasping in clutter via singulation," in 2019 International Conference on Robotics and Automation (ICRA), Montreal, QC, Canada, 2019, pp. 1596-1600.

[2] L. Chang, J. R. Smith, and D. Fox, “Interactive singulation of objects from a pile,” in 2012 IEEE International Conference on Robotics and Automation. IEEE, 2012, pp. 3875–3882.

[3] M. Danielczuk, J. Mahler, C. Correa, and K. Goldberg, “Linear push policies to increase grasp access for robot bin picking,” in 2018 IEEE 14th International Conference on Automation Science and Engineering (CASE). IEEE, 2018, pp. 1249–1256.

Author Response

Attached is the reply to the reviewer's comment

Round 3

Reviewer 1 Report

This paper deals with the singulation of identical circular disks with two non-prenhesile actions, pushing and hitting. Pushing is used to singulate two contacting disks and hitting for breaking a cluster of disks. In the revised manuscript, more detailed explanations are included, however, the reviewer has the following concerns:

1) Although the authors claim that they have revised the paper in order to be more readable, there still exists parts of the papers that are very difficlut to follow. Related work should be a seperate section in order to highlight the benefits of the proposed approach with sota. As for the second contribution is concerned, this is still vague. The authors write 'introduce initial direction'? Do you mean that you design the way the primitives are executed?
Overall, there are still many parts that need to be rewritten in the revised manuscript.

2) The authors claim that they cannot compare their work with other approaches. However, I think that a qualitative comparison with an analytic baseline should be done in order to highlight the importance of the proposed approach. For example, a naive baseline would be to push each disk towards free space.

3) Experiments with a more diverse set of objects are not conducted. At least the authors should conduct more experiments to improve the power of the statistical analysis. The statistics in Figure 13 are computed from only 3 experiments.

4) An interesting experiment would be to compare the performance of the proposed approach in assymetric formations and billiard formations.

5) Adding the number of pushes as performance measure would be more understable from the time to singulation.

Minor:
- Refs 4 and 8 are the same

Author Response

A file is attached. 

Thanks for the valuable comments.
